# Prospective observational study of SARS-CoV-2 infection, transmission and immunity in a cohort of households in Liverpool City Region, UK (COVID-LIV): a study protocol

Wega Setiabudi [1], Daniel Hungerford [1,2,3] Krishanthi Subramaniam,[1] Natasha Marcella Vaselli,[1] Victoria E Shaw,[4] Moon Wilton,[5] Roberto Vivancos,[2,3,6] Stephen Aston [1] Gareth Platt,[1] Tracy Moitt,[7] Ashley P Jones,[7] Mark Gabbay,[2,8] Iain Buchan [3,9] Enitan D Carrol,[1] Miren Iturriza-Gomara,[1,3] Tom Solomon,[1,2,10] William Greenhalf,[4] Dean J Naisbitt,[11] Emily R Adams,[2,12] Nigel A Cunliffe,[1,3] Lance Turtle,[1,2] Neil French,[1,2] on behalf of the COVID-LIV Study Group

► Prepublication history and supplemental material for this paper is available online. To view these files, please visit the journal online (http://dx.doi.org/10.1136/bmjopen-2020-048317).

WS and DH contributed equally.

WS and DH are joint first authors.

For numbered affiliations see end of article.

**Correspondence to**
Professor Neil French;
french@liverpool.ac.uk

## ABSTRACT

**Introduction** The emergence and rapid spread of COVID-19 have caused widespread and catastrophic public health and economic impact, requiring governments to restrict societal activity to reduce the spread of the disease. The role of household transmission in the population spread of SARS-CoV-2, and of host immunity in limiting transmission, is poorly understood. This paper describes a protocol for a prospective observational study of a cohort of households in Liverpool City Region, UK, which addresses the transmission of SARS-CoV-2 between household members and how immunological response to the infection changes over time.

**Methods and analysis** Households in the Liverpool City Region, in which members have not previously tested positive for SARS-CoV-2 with a nucleic acid amplification test, are followed up for an initial period of 12 weeks. Participants are asked to provide weekly self-throat and nasal swabs and record their activity and presence of symptoms. Incidence of infection and household secondary attack rates of COVID-19 are measured. Transmission of SARS-CoV-2 will be investigated against a range of demographic and behavioural variables. Blood and faecal samples are collected at several time points to evaluate immune responses to SARS-CoV-2 infection and prevalence and risk factors for faecal shedding of SARS-CoV-2, respectively.

**Ethics and dissemination** The study has received approval from the National Health Service Research Ethics Committee; REC Reference: 20/HRA/2297, IRAS Number: 283 464. Results will be disseminated through scientific conferences and peer-reviewed open access publications. A report of the findings will also be shared with participants. The study will quantify the scale and determinants of household transmission of SARS-CoV-2. Additionally, immunological responses before and during the different stages of infection will be analysed, adding to the understanding of the range of immunological response by infection severity.

## Strengths and limitations of this study

► Liverpool Household COVID-19 Cohort Study is a prospective cohort study of households that aims to represent the socioeconomic profile of the Liverpool City Region population, which enables the determination of risk factors of SARS-CoV-2 transmission while minimising recall bias.

► This household-based study will identify paucisymptomatic and asymptomatic COVID-19 cases, thus allowing the measurement of their contribution to transmission.

► The longitudinal nature of the study enables the capture of subjects before they test positive for COVID-19, which provides a preinfection and postinfection time point to evaluate changes to the host immune response.

► Limitations include the relatively small sample size and repeated self-sampling, which may lead to diagnostic inconsistencies.

► Participation bias by those most engaged with COVID-19 and disease control may theoretically result in an unrepresentative study cohort.

## INTRODUCTION

Within months of the first reports of a novel respiratory disease in Wuhan, China, in December 2019, COVID-19 has been declared a global pandemic with devastating impacts.[1] The disease caused by the SARS-CoV-2 has reached 46.8 million cases and 1.2 million deaths globally as of 3 November 2020, although the real number is likely to be much higher.[2] With limited evidence of effective prophylactic treatment and prior to widespread availability of vaccination, countries

around the world have been forced to implement various forms of restriction to individual movement in order to control the transmission of SARS-CoV-2.[3–5] Although the effectiveness of such measures in controlling viral transmission comes at the cost of disruption in socioeconomic activity and mental well-being, the impact from uncontrolled transmission could cause the loss of millions of lives and the potential collapse of health systems.[6–9]

Understanding the pattern of community transmission is essential to inform approaches to contain the spread of SARS-CoV-2. The role of transmission between household members is believed to have a significant role in the spread of the disease, where the secondary attack rate (SAR) is estimated to be 16.6%.[10] This is reflected in the current UK government guideline where members of the household of a confirmed case are required to self-isolate for 10 days.[11] Despite this, the limited availability of long-term prospective cohort studies means that further exploration of how SARS-CoV-2 transmits within households, and a better understanding of how immune responses develop over time is urgently needed.[10 12]

In October 2020, the Liverpool City Region became the first area in England, UK, to be placed in the highest level of regional restriction after experiencing one of the highest rate of infection in the country.[13] The region was also chosen as the site for the pilot asymptomatic mass testing due to its high infection rate during the second national lockdown.[14] The transmission characteristic of the Liverpool City Region could be explored further through a community study of the virus transmission. These data would aid understanding of the transmission dynamics of SARS-CoV-2 that may be beneficial in informing public health interventions.

The Liverpool Household COVID-19 Cohort Study (COVID-LIV) is a prospective observational study of households in the Liverpool City Region. As a household-based study, COVID-LIV will capture paucisymptomatic and asymptomatic COVID-19 cases. This allows measurement of the role of different disease manifestation of COVID-19 cases in the transmission of SARS-CoV-2 between household members. In addition, the prospective nature of the study allows characterisation of the immune response to SARS-CoV-2 at different stages of the infection and determine the durability of the response.

## STUDY AIMS

Among households in the Liverpool City Region, the primary aim is to understand household associated SARS-CoV-2 transmission. This aim will be achieved by:

1. Measurement of household COVID-19 incidence and SARs.
2. Identification of the determinants of transmission of SARS-CoV-2.
3. Estimation of the contribution of paucisymptomatic and asymptomatic infection to the spread of SARS-CoV-2.

Secondary aims are:

1. Measure family member contact patterns and the relationship to household structure.
2. Describe the clinical phenotype of mild COVID-19 cases.
3. Undertake sequence of SARS-CoV-2.
4. Characterise the immune response in mild COVID-19 cases.
5. Characterise the immune response of exposed household contacts with no subsequent detection of confirmed infection.
6. Investigate the prevalence of household faecal shedding of SARS-CoV-2.

## METHODS AND ANALYSIS

### Design

COVID-LIV is a prospective observational cohort study of households in the Liverpool City Region, UK. Cohorts are followed up for an initial period of 12 weeks and then up to 3 years, observing the incidence of household transmission of SARS-CoV-2 and characterising changes in the immune response over time. For 12 weeks, all household members are requested to perform weekly self-administered throat and nasal swabbing, along with the collection of blood and other clinical samples at different time points of the study. The study started in July 2020 and is expected to continue until September 2023.

There are social science studies linked to this household study, including longitudinal surveys of all these households focusing on the impacts of the pandemic on the residents included in this research. In addition, there will be in-depth qualitative interviews at baseline and 3 months with a purposive sample of these households, focusing on risk perception beliefs and actions.

### Study population

Households are recruited from the large metropolitan Liverpool City Region in North West England, UK. The Liverpool City Region comprises six local authorities and has a population of over 1.5 million people. Almost 50% of its population are categorised as living in the 20% most deprived areas of England.[15]

### Recruitment procedure

Households were recruited from the established Liverpool household survey undertaken by the National Institute for Health Research Collaboration for Leadership in Applied Health Research and Care (CLAHRC, now ARC).[16] The established study framework contains over 7000 households, representing a spectrum of demographic and socioeconomic characteristics. The initial selection process was undertaken by ARC team members with appropriate permission to access the original survey data. Individuals who indicated a willingness to be recontacted for further research participation were identified and contacted by the COVID-LIV study team about their potential participation in this study.

To supplement the number of recruits, other methods to reach out to potential participants were used, which

include text messages sent from local general practitioner (GP) surgeries to their patients containing information about the study and sharing study information through the University of Liverpool media, local media outlets and the social media of the researchers and stakeholder organisations. Interested households that contacted the study team are recruited if they fulfil the study criteria as follows:

### Inclusion criteria
1. All people in the household willing to take part.
2. At least one adult within the household must speak English and willing to translate.
3. Have provided informed verbal and written consent or personal legal consent for those lacking capacity.
4. Ability and willingness to undertake self-swabbing.
5. Intention to be resident for at least 6 months within their current household, except for students, military personnel and other professions who may have to move away from home for purposes of study or employment

### Exclusion criteria
1. Contraindication to throat and nasal swab or blood sampling.
2. One or more members of the household have had a proven COVID-19 test (positive PCR test for SARS CoV-2).
3. No members of the household can speak English.

A household comprises those individuals who reside at the household at the date of contact, even if they do not believe this is their primary residence and are intending to stay for at least 6 months beyond the date of enrolment and first sampling. Regularly attending persons such as carer and cleaner are classified as attendees and are asked to participate in the study, although their participation status does not affect the eligibility of the household. A participant aged 16 years or above is considered an adult.

### Participant pathway
This section provides the pathway details of study participant from enrolment up to the end of the study (figure 1).

### Consent procedure
The consent process consists of two phases: an introductory communication, followed by a visit to establish consent and sampling. At the first phase, potential households from the list of contacts from ARC will be contacted by phone or email wherever possible for ease and speed of communication and to minimise transmission risk. The same method of communication applies to participants that directly contacted the study team in response to advertisement through GP surgeries and other forms of media communication. During this phase, potential participants are given a brief explanation about the study, access to information on the study website is confirmed and any queries are answered.

Following initial contact and expression of interest by the household, a visit arrangement by research nurses is made. During the visit, where every household member is expected to be present, printed information sheets are provided along with further discussions on the purpose of the study and procedures required (see online supplemental appendix file 1). Written informed consent is expected to be provided for each member of the household. In addition to the parent or guardian consent form, children are provided with age-relevant information sheets; assent is obtained if the child is aged 8–15 years old and deemed capable of assessing the study documentation provided.

### Baseline visit
After written consent forms have been acquired from all household members, a baseline visit date is arranged. The visit is done on the day the consent forms are signed, or another date is arranged if necessary. During the baseline visit, the research nurses collect throat and nasal swab, blood samples (or finger prick sample if not suitable for venepuncture), nasal mucosal sample and saliva sample from all adult participants. Only a finger prick sample and saliva sample are collected from children. The research nurses also train the participants on how to perform throat and nasal swabbing themselves. Instruction on the procedure of self-swabbing is given to each household, along with the swab kits for the following weeks.

### The first 12 weeks of participation
Participants are instructed to perform self-throat and nasal swab every week for a total of 12 weeks after enrolment; samples are collected by a courier. A questionnaire is sent each week through email or phone call if no email address is provided, requesting information about the participant's health condition and activities from the past 7 days. Participants are also asked to report any COVID-19 test done outside the study system, both during and after the initial 12 weeks of self-swabbing. Optional stool samples are collected from consenting participants at weeks 6–8 and 12–14 from enrolment.

### Positive swab and result notification
If a positive SARS-CoV-2 swab result occurs during the first 12 weeks of self-throat and nasal swabbing, participants are informed of the result within 72 hours of sample receipt at the laboratory. Positive case details are passed to the National Health Service (NHS) test and trace according to Public Health England statutory requirement. On notification, participants are given information on self-isolation and are provided with other relevant guidelines from the UK government and the NHS. The participant's GP is also informed, and additional clinical advice is available from the infectious diseases team at Liverpool University Hospitals NHS Foundation Trust or Alder Hey Children's Hospital NHS Foundation Trust if deemed necessary by the study clinical team.

Following notification, a household visit is arranged to obtain additional samples of blood (or finger prick), throat and nasal swab, nasal mucosal swab and saliva from the positive case and other household members. The visit

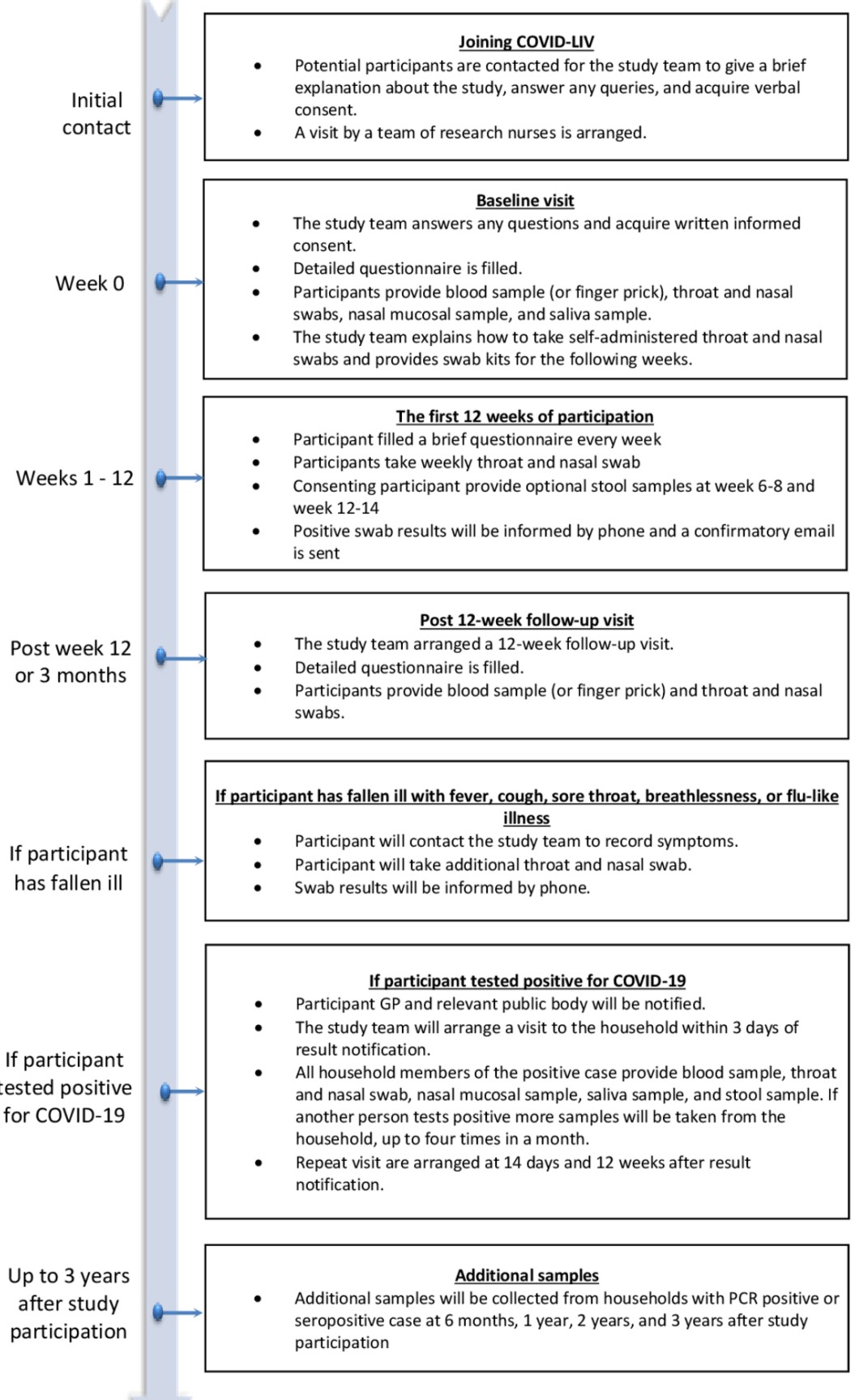

**Figure 1** Participant pathway. COVID-LIV, Liverpool Household COVID-19 Cohort Study; GP, general practitioner.

**Table 1** Collection of clinical samples for the COVID-LIV study

| Timepoint | Baseline/ enrolment | Week 1–12 (day 8, day 15 and so on) | Weeks 6–8 and 12–14 | Any time in weeks 0–12 | Week 12 follow-up | Week 12, month 6, years 1, 2 and 3 |
|---|---|---|---|---|---|---|
| Participant group | All participants | All participants | All participants* | PCR swab positive participants/household members of swab positive participants† | PCR swab negative participants | Seropositive/PCR swab positive participants and their household members |
| Throat and nasal swab | × | × | | × | | × |
| Blood samples | × | | | × | × | × |
| Stool Sample | | | ×* | × | | |
| Nasal mucosal sample | ×‡ | | | ×‡ | | ×‡ |
| Saliva sample | × | | | × | | × |

*Optional – additional consent required.
†Samples taken at day 3 and day 14 after PCR swab positive test.
‡Adult participant only.
COVID-LIV, Liverpool Household COVID-19 Cohort Study.

is expected to be done within 3 days following result notification; repeat visits are arranged at 14 days and 12 weeks after result notification. If a participant has more than one positive PCR swab result during the course of the study, this will trigger restart of the additional sampling schedule if more than 6 weeks have elapsed from the first positive test in order to identify reinfection.

### Follow-up visits

The first follow-up visit is performed at 12 weeks after enrolment for households with and without COVID-19 cases. Samples of blood (or finger prick) are collected from households with no positive case, while samples of blood (or finger prick), throat and nasal swab, nasal mucosal swab and saliva are collected from households with at least one positive case. Repeat visits to households with a history of a positive throat and nasal swab, or seropositivity at baseline or at 12-week follow-up, will be conducted at 6 months, 1 year, 2 years and 3 years postenrolment.

### Clinical sample and laboratory investigation

The following section provides more detail on the clinical samples that are obtained at different time points during the study (table 1).

### Throat and nasal swab

A combined throat and nasal swab are taken for detection of SARS-CoV-2 at baseline by the participant under the guidance from the research nurses. Swabs are then taken by the participants at home and collected on a weekly basis for 12 weeks. This sample is collected using nylon flocked dry swabs placed into plastic tubes and transported to the laboratory to be tested within 72 hours. Participants are asked to perform self-swabbing on the night before

or in the morning of collection day, where samples are then collected within 12–24 hours. Samples are placed inside a specimen bag and specimen cardboard box and stored at ambient temperature until collection and during transport. Swabs are processed for RNA extraction (Zymo Research) and qPCR (Primer Design Novacyt). Remaining Amies medium and extracted nucleic acid will be stored for future virology research.

### Virus sequencing

Nucleic acid from positive throat and nasal swabs (and a small number of negative swabs as controls) will be transferred to the Centre for Genomic Research, University of Liverpool for SARS-CoV-2 whole-genome sequencing using the nanopore technology and the ARTIC network protocol.[17 18]

### Blood sample

Up to 60 mL of whole blood are collected from each adult participant (or finger prick sample, if unsuitable for venepuncture). Children under the age of 16 years will have finger prick and blood spot collection rather than venepuncture and a smaller amount of blood collected.

The baseline and 12-week follow-up blood samples will be used to determine the prevalence of exposure to SARS-CoV-2 at a certain point of the epidemic. These data will be used to supplement virology data to maximise the identification of SARS-CoV-2 exposure.

Peripheral blood mononuclear cells (PBMCs) will be isolated from different time points of infection using Ficoll density centrifugation. Briefly, blood collected from sodium heparin tubes will be placed on a Ficoll cushion and centrifuged to retrieve PBMCs. Cells will then be washed and frozen down in 90% fetal bovine

serum and 10% dimethyl sulfoxide for downstream assays.

### Antibody responses

The antibody response will be measured over time at baseline, 12 weeks, 6 months, and 1, 2 and 3 years postinfection by ELISA, pseudovirus neutralisation and SARS-CoV-2 neutralisation in a subset. The proportion of participants positive at each time point will be determined, and the magnitude of antibody titres will be measured. If positive cases are detected, neutralising antibody titres will be measured, and virus isolation will be attempted allowing testing of the serum neutralising capacity against the actual virus infecting the participant. Mucosal antibody and cytokine responses will be tested. These experiments will determine whether serum antibody measurements correlate with mucosal antibody and whether either is an adequate correlate of immunity. Parallel samples from household contacts (who are highly likely to be exposed) will also be collected and studied in order to determine what, if any, factors protect against the acquisition of infection or correlate with sterilising immunity.

### T cell responses

Antigen-specific responses will be measured following ex vivo stimulation with SARS-CoV-2 peptide pools. PBMCs isolated at the various time points will be stimulated with various peptide pools and intracellular cytokine stain, and activation marker assays will be performed to characterise the SARS-CoV-2 T cell responses. Single-cell RNA-seq assays will also be done to explore the breadth of the T cell response to determine qualitative differences in the T cell repertoire. Where sample allows, T cell epitopes will be mapped using a synthetic peptide library and tested for cross-reactivity against common cold coronaviruses.

### Innate response

Whole blood stored in RNA stabilisation solution (tempus tubes) will be subjected to RNA isolation and sequencing to characterise the innate immune response. These data will inform and refine the above experiments and have the potential to be related to the ISARIC 4C dataset (hospitalised severe cases) as a mild disease group.[19]

### Genomic testing

Human leucocyte antigen typing will be undertaken along with characterisation of other important immune mediating characteristics, such as angiotensin receptor 2.

### Stool sample

Stool samples are collected from adult participants who test positive from a PCR swab and from their consenting household contacts at approximately 3 and 14 days after confirmation of a positive PCR. Samples are transported to the University of Liverpool where they are frozen down for downstream assays, including for SARS-CoV-2 sequencing. Additionally, optional stool samples are requested from all participants at two time points from

their enrolment at approximately week 6–8 and week 12–14.

### Nasal mucosal and saliva sample

At the baseline visit, nasal mucosal and saliva samples are collected from adult participants and all participants, including children, respectively. The nasal mucosal sample is collected using synthetic absorptive matrix strips, and saliva sample is collected using ORACOL+ (Malvern Medical Developments), both are collected for antibody analyses. Additional samples are also collected from adult participants who subsequently tested positive from PCR swab and their household contacts.

## Outcome measures

### Primary endpoints

1. Incidence of paucisymptomatic and asymptomatic SARS-CoV-2 infections index cases, including the prevalence of infection or past infection at baseline serology status.
2. Incidence of secondary household cases.
3. Risk factors for household transmission.

### Secondary endpoints

1. Analysis of household contact patterns.
2. Description of clinical phenotypes of the index cases.
3. Genomic characterisation of SARS-CoV-2.
4. Characterisation of the immune response in index cases and exposed household contacts.
5. Prevalence of SARS-CoV-2 household faecal shedding.

## Data analysis

The results of the analyses will be reported according to the Strengthening the Reporting of Observational Studies in Epidemiology guidelines.[20] This will include a descriptive analysis of households, paucisymptomatic and asymptomatic primary household index cases and secondary household cases.

Environmental, demographic and behavioural risk factors for secondary transmission among household contacts of symptomatic and laboratory-confirmed cases of COVID-19 will be investigated. Cases in households will be ordered by date of symptom onset. The first symptomatic COVID-19 case in the household will be classified as the probable household index case. Secondary COVID-19 cases will be defined as any COVID-19 case with an onset of illness within 7 days following the onset of the index case.

The primary attack rate will be calculated as the number of households with a primary case divided by the total number of households in the study. The household SAR will be calculated from the number of households with at least one secondary COVID-19 case divided by the total number of households at risk. The individual household members attack rate will be calculated from the number of household members with secondary COVID-19 illness divided by the total number of household members at risk.

Index cases will be described in relation to demographics, employment and contact history. A risk factor analysis will be undertaken to investigate variables associated with secondary attack cases within households. Risk factors will include data on contacts, viral load measurements, household characteristics and other variables that emerge in external reports or literature that may be linked with transmission.

Data and statistical analysis of serology and other immunological parameters will be done using GraphPad Prism, FlowJo V.10, R and other bioinformatics software.

### Sample size

The study should be regarded as exploratory. The initial constraints on sample size are primarily access to testing on a weekly basis. Referring to the current data on the SAR of 10.5%–45% of contacts with a HR of 1.5 or 2.0, using a single sample Cox proportional hazard model with 80% power and 10% study withdrawal, we propose an initial sample size of 300 households, which will contain approximately 1000 individuals.[21–23]

### Patient and public involvement (PPI)

The protocol has been reviewed by the PPI committee of the Institute of Infection, Veterinary, & Ecological Sciences, University of Liverpool. The study design, participant acceptability and perceptions have been reviewed and discussed. The necessary speed to get this work up and underway has prevented a more standard input from the PPI group. Test results will be reported to participants in plain language.

### ETHICS AND DISSEMINATION

The study has received approval from the NHS Research Ethics Committee; REC Reference: 20/HRA/2297, IRAS Number: 283 464. Protocol amendments have been and will be reported to the Research Ethics Committee, as will any serious adverse events. The study participants are informed that all data collected are for research purposes only and that they have the right to withdraw from the study at any time.

### Project governance

The study is coordinated by the Liverpool Clinical Trials Centre, University of Liverpool. A study steering group has been established to enable effective achievement of the project objectives. The steering group includes representatives from academia, public health and lay membership.

### Dissemination of research findings

The findings will be presented at professional and scientific conferences. The results will also be published in peer-review publications and, if appropriate, published first as preprints to enable a timely public health response to COVID-19. Interim and final reports will be submitted to the funders and the steering group. We also work with our institute PPI panel to identify and produce materials to disseminate to the general public, including study participants.

### DISCUSSION

COVID-LIV will demonstrate the role of household transmission of SARS-CoV-2 in a cohort of households in the Liverpool City Region, UK. By observing households with no apparent previous infection of COVID-19, it is hoped that the incidence, determinants of transmission and contribution of paucisymptomatic and asymptomatic cases can be described, filling a knowledge gap in how the disease transmits within the population in the Liverpool City Region. Characterising immune responses in a cohort of mild infections will provide a better understanding of how natural infection alters immune parameters over time, allowing a better understanding of immunity against COVID-19 infection that may help inform vaccine development and delivery.

### Strengths

COVID-LIV aimed to recruit a cohort of households across a representative range of socioeconomic status in the Liverpool City Region. The household cohort allows for identification of paucisymptomatic and asymptomatic COVID-19 cases, which will provide a better representation of the impact of COVID-19 in the community and extent of transmission through the sampling of high probability exposed household members. The prospective nature of the study allows the determination of a true incidence rate and risk factors for SARS-CoV-2 transmission with less recall bias. The longitudinal study design enables the analysis of the immunological response and faecal shedding of SARS-CoV-2 during different stages of the disease. It also allows observation of the natural progression of mild cases from a preinfection stage sample collection to allow the interrogation of T cell repertoires and their association with acute infection.

### Limitations

The cohort households may be biased by those that are most engaged with COVID-19 and disease control, leading to a low level of secondary infections as participating individuals are more likely to take precautions against transmission. Low level of detectable infections may also be observed due to the study observation across different seasonality time points. Reliance on repeated self-sampling may lead to diagnostic inconsistencies, although instructions were given, and techniques were carefully assessed by the research nurses during the initial baseline visit. Exclusion of non-English-speaking families may exclude potential high-risk households resulting in under detection of incidence rate and more severe cases.

**Author affiliations**
[1]Department of Clinical Infection Microbiology and Immunology, Institute of Infection, Veterinary & Ecological Sciences, University of Liverpool, Liverpool, UK
[2]NIHR Health Protection Research Unit in Emerging and Zoonotic Infections, University of Liverpool, Liverpool, UK

[3]NIHR Health Protection Research Unit in Gastrointestinal Infections, University of Liverpool, Liverpool, UK

[4]Liverpool Experimental Cancer Medicines Centre, Institute of Systems, Molecular and Integrative Biology, University of Liverpool, Liverpool, UK

[5]Department of Psychology, Institute of Population Health, University of Liverpool, Liverpool, UK

[6]Field Epidemiology North West, Field Service, National Infection Service, Public Health England, London, UK

[7]Liverpool Clinical Trial Centre, University of Liverpool, Liverpool, UK

[8]Department of Primary Care and Mental Health, Institute of Population Health, University of Liverpool, Liverpool, UK

[9]Department of Public Health and Policy, Institute of Population Health, University of Liverpool, Liverpool, UK

[10]Department of Neurology, Walton Centre, NHS Foundation Trust, Liverpool, UK

[11]Department of Molecular and Clinical Pharmacology, Institute of Translational Medicine, University of Liverpool, Liverpool, UK

[12]Department of Tropical Disease Biology, Liverpool School of Tropical Medicine, Liverpool, UK

**Acknowledgements** The authors would like to thank and appreciate all the participants in the study for their invaluable contribution and the external advisory panel members: Jonathan M Read, Antonia Ho and Cliona McDowell.

**Collaborators** The following are members of the COVID-LIV Study Group: Principal investigator: Neil French. Study investigators: Lance Turtle; Dan Hungerford; Krishanthi Subramaniam; Roberto Vivancos; Mark Gabbay; Iain Buchan; Enitan D Carrol; Miren Iturriza-Gomara; Tom Solomon; Nigel A Cunliffe; Emily R Adams and Carrol Gamble. Lay members: Lynnette Crossley and Neil Joseph. Fieldwork team: Wega Setiabudi; Natasha Marcella Vaselli; Moon Wilton; Lee D Troughton; Samantha Kilada; Katharine Abba; Victoria Simpson; John S P Tulloch; Lynsey Goodwin; Rachael Daws; Shiva S Forootan; Susan Dobson; Rachel Press; Vida Spaine; Lesley Hands; Kate Bradfield and Carol McNally. Project management: Tracy Moitt; Silviya Balabanova; Chloe Donohue; Lynsey Finnetty and Laura Marsh. Clinical and laboratory team: William Greenhalf; Dean J Naisbitt; Victoria E Shaw; Stephen Aston; Gareth Platt; Paul J Thomson; Monday Ogese; Sean Hammond; Kareena Adair; Liam Farrell; Joshua Gardner; Kanoot Jaruthamsophon; Serat-E Ali; Adam Lister; Laura Booth; Milton Ashworth; Katie Bullock; Benjamin W A Catterall; Terry Foster; Lara Lavelle-Langham; Joanna Middleton; William Reynolds; Emily Cass; Alejandra Doce Carracedo; Lianne Davies; Lisa Flaherty; Melanie Oates; Nicole Maziere; Jennifer Lloyd; Christopher Jones; Hannah Massey; Anthony Holmes; Nicola Carlucci; Vanessa Brammah; Yasmyn Ramos; Daniel Allen; Jane Armstrong; Debbie Howarth; Eve Wilcock; Jena Lowe; Jayne Jones; Paula Wright; Iain Slack; Simone McLaughlin; Jessica Mason; Thomas Edwards; Claudia McKeown; Elysse Hendrick; Chris Williams; Rachel Byrne; Kate Buist; Gala Garrod; and Sophie Owen. Statisticians: Ashley P Jones and Efstathia Gkioni.

**Contributors** NF, NAC, DH, LT, MI-G conceived of the study. DH, KS, NF, NAC, LT, MI-G, TS, SA, IB, MG, RV, MW, NMV, WS, EDC and ERA initiated study design and protocol development. GP, VES, WG, DJN and TM helped with study implementation. DH, NF and APJ provide statistical expertise in statistical design and have produced the analysis plan. WS and DH drafted the manuscript. All authors contributed to the refinement of the study protocol.

**Funding** This study is cofunded by the National Institute for Health Research Health Protection Research Unit (NIHR HPRU) in Gastrointestinal Infections, a partnership between Public Health England (PHE), the University of Liverpool and the University of Warwick; the NIHR HPRU in Emerging and Zoonotic Infections, a partnership between PHE, the University of Liverpool in collaboration with the Liverpool School of Tropical Medicine and the University of Oxford; the Centre of Excellence in Infectious Disease Research (CEIDR); and the Alder Hey Charity. Grant number for the fundings is not applicable. NF is funded by the NIHR HPRU in Emerging and Zoonotic Infections, the CEIDR and the Alder Hey Charity. We also acknowledge the support of Liverpool Health Partners and the Liverpool-Malawi-Covid-19 Consortium. This research was funded in whole, or in part, by a Wellcome Trust fellowship awarded to LT (205228/Z/16/Z). For the purpose of open access, the author has applied a CC BY public copyright licence to any Author Accepted Manuscript version arising from this submission. LT is also supported by the NIHR HPRU in Emerging and Zoonotic Infections (NIHR200907) at University of Liverpool in partnership with PHE, in collaboration with Liverpool School of Tropical Medicine and the University of Oxford. LT is based at University of Liverpool. WS is funded by the Ministry of Finance, the Republic of Indonesia through the Indonesia Endowment Fund for Education (Lembaga Pengelola Dana Pendidikan or LPDP) scholarship for doctoral study (201807220413052). DH is funded by an NIHR Postdoctoral Fellowship (PDF-2018-11-ST2-006). KS is funded by a HEFCE-funded University of Liverpool Tenure Track Fellowship. EDC acknowledges funding from the NIHR i4i Programme (II-LA-0216-20002), HTA Programme (15/188/42, 17/136/13), EME Programme (NIHR129960) and H2020 (Project No. 848196). MG is part-funded by the NIHR Applied Research Collaboration North West Coast.

**Disclaimer** The views expressed are those of the author(s) and not necessarily those of the NHS, the NIHR, the Department of Health and Social Care, PHE or other funding bodies.

**Competing interests** MW is funded under a grant from Astra Zeneca and University of Liverpool for an unrelated project.

**Patient consent for publication** Not required.

**Provenance and peer review** Not commissioned; externally peer reviewed.

**ORCID iDs**
Wega Setiabudi http://orcid.org/0000-0003-2510-0318
Daniel Hungerford http://orcid.org/0000-0002-9770-0163
Stephen Aston http://orcid.org/0000-0002-0701-8364
Iain Buchan http://orcid.org/0000-0003-3392-1650

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
