## [Reviewer comments · BMJ Open]

ARTICLE DETAILS

TITLE (PROVISIONAL)	Prospective observational study of SARS-CoV-2 infection, transmission, and immunity in a cohort of households in Liverpool City Region, UK (COVID-LIV): a study protocol
AUTHORS	Setiabudi, Wega; Hungerford, Daniel; Subramaniam, Krishanthi; Vaselli, Natasha; Shaw, Victoria E.; Wilton, Moon; Vivancos, Roberto; Aston, Stephen; Platt, Gareth; Moitt, Tracy; Jones, Ashley P.; Gabbay, Mark; Buchan, Iain; Carrol, Enitan; Iturriza-Gomara, Miren; Solomon, Tom; Greenhalf, William; Naisbitt, Dean J.; Adams, Emily; Cunliffe, Nigel; Turtle, Lance; French, Neil

VERSION 1 – REVIEW

REVIEWER	Bruce S. Gillis, MD MPH EpicGenetics USA
REVIEW RETURNED	01-Feb-2021

GENERAL COMMENTS	Comprehensive approach, though limiting the study to 12 weeks is probably inadequate. I recommend a longer survey period of time.
---

REVIEWER	Zefferino Roberto University of Foggia Italy
REVIEW RETURNED	03-Feb-2021

GENERAL COMMENTS	The study is designed very well and it has an undoubt social value, as well as scientific value. Moreover it is possible, in my opinion, to meliorate it through some changes. a) the most important limitation is related to “to perform self-throat ...”; we know that every individual has different ability to learn and you could report false negative, due to inappropriate procedure; I think that you should find a different solution (i.e. every two week you could perform double swabs (contemporary self and nurse collect)); b) you will collect salivary sample using “oral fluid collection device”, please you should specify better the type of salivary sampler and what you will measure in the saliva c) At week 12 it is not correct to collect only blood samples to PCR swab negative subject, in my opinion, you should evaluate blood, nasal mucosal, saliva sample and possibly stool, as you do with PCR swab positive subject. d) In your protocol you specified that the swabs will be transported to the laboratory and tested within 72 h but you should indicate where samples are stored before the transport and above all the temperature (to store for 72 h at 5 ° C or -18 ° C may not be the
--

	right solution), you should also indicate the temperature during the transport. e) I would like to add that it could also be useful evaluating the cytokines in the serum in order to determinate the Th1/Th2 ratio in each subject, then verifying if it can correlate with different ability to respond to contact with positive case. f) During baseline visit it could be useful to collect subject's medical history regarding to previous vaccines, reporting the type.
--	--

VERSION 1 – AUTHOR RESPONSE

Reviewer #1:

Comprehensive approach, though limiting the study to 12 weeks is probably inadequate. I recommend a longer survey period of time.

We thank the reviewer for this comment. To clarify, active self-swabbing are performed for 12 weeks and any household with a positive case or who are seropositive at baseline will be followed-up at regular intervals up to three years. We also ask participants to report any tests done during and outside of the 12-week self-swabbing period as Liverpool City Region has a comprehensive symptomatic and asymptomatic COVID-19 testing system.

We have added the following to page 9, line 14 (reference to peer review copy format) in the manuscript for clarification: "Participants are also asked to report any COVID-19 test done outside the study system, both during and after the initial 12 weeks of self-swabbing."

Reviewer #2:

a) The most important limitation is related to "to perform self-throat ..."; we know that every individual has different ability to learn and you could report false negative, due to inappropriate procedure; I think that you should find a different solution (i.e. every two week you could perform double swabs (contemporary self and nurse collect));

We agree this is a limitation of self-swabbing and indicate this on page 14, line 29 (reference to peer review copy format) of the protocol. However, unlike public mass testing platforms, we have research nurses provide a comprehensive tutorial on how to perform self-swabbing at the baseline visit. Printed instructions on how to perform self-swabbing that has been reviewed by the PPI committee and ethics board is also given to each household. Due to resource limitations and risk management (minimising contact as much as possible), we are unfortunately unable to perform the suggested solution.

b) You will collect salivary sample using "oral fluid collection device", please you should specify better the type of salivary sampler and what you will measure in the saliva

We have added the following to page 12, line 7 (reference to peer review copy format) in the manuscript for clarification: "... ORACOL+ (Malvern Medical Developments, UK) ...". Both the nasal mucosal and saliva samples are collected for antibody analyses, as stated on page 12, line 7.

c) At week 12 it is not correct to collect only blood samples to PCR swab negative subject, in my opinion, you should evaluate blood, nasal mucosal, saliva sample and possibly stool, as you do with PCR swab positive subject.

We agree that ideally, a more comprehensive sampling should have been collected from every participant at week 12 of follow up. However, due to logistical limitations, this was not feasible. We do collect blood, nasal mucosal, saliva and stool samples from every household member that had a positive case and by so doing will have a sub-sample of test negatives to study.

d) In your protocol you specified that the swabs will be transported to the laboratory and tested within 72 h but you should indicate where samples are stored before the transport and above all the temperature (to store for 72 h at 5 ° C or -18 ° C may not be the right solution), you should also indicate the temperature during the transport.

We ask participants to perform self-swabbing on the night before or in the morning of the collection date. To avoid storage concerns, we use a flocked swab placed inside a dry tube. We ask each participant to put their swab tube along with their household members' inside a specimen bag which is then placed inside a specimen cardboard box and sealed with a security sticker that is provided. The samples are at ambient temperature until they are transported to the laboratory.

We have added the following to page 10, line 42 (reference to peer review copy format) in the manuscript for clarification: "Participants are asked to perform self-swabbing on the night before or in the morning of collection day, where samples are then collected within 12 to 24 hours. Samples are placed inside a specimen bag and specimen cardboard box and stored at ambient temperature until collection and during transport."

e) I would like to add that it could also be useful evaluating the cytokines in the serum in order to determinate the Th1/Th2 ratio in each subject, then verifying if it can correlate with different ability to respond to contact with positive case.

We thank the reviewer for this comment and agree that evaluating serum cytokine levels will be important to understanding host responses. Studies evaluating the cytokine and chemokine profiles in cohorts with mild and severe SARS-Cov-2 infections do exist (PubMed ID: 32855555, 33277181, 32896310). However, as our study also has a baseline component, we agree this is worthy of exploration, although we do not feel that we need to include this level of detail in the clinical protocol.

f) During baseline visit it could be useful to collect subject's medical history regarding to previous vaccines, reporting the type.

As is stated in one of the options of the consent form, we have the option to contact the participant's general practitioner for acquiring information from their health record, including vaccine history. This approach should provide a more robust approach and avoid recall bias.

VERSION 2 – REVIEW

REVIEWER	Roberto Zefferino University of Foggia Italy
REVIEW RETURNED	28-Feb-2021
GENERAL COMMENTS	The manuscript was meliorated by Authors and , in my opinion, can be published.